# Functional Analysis of KAP1/TRIM28 Requirements for HIV-1 Transcription Activation

**DOI:** 10.3390/v16010116

**Published:** 2024-01-13

**Authors:** Keyera Randolph, Usman Hyder, Ashwini Challa, Erick Perez, Iván D’Orso

**Affiliations:** Department of Microbiology, The University of Texas Southwestern Medical Center, Dallas, TX 75390, USA; keyera.randolph@utsouthwestern.edu (K.R.); usman.hyder@utsouthwestern.edu (U.H.);

**Keywords:** KAP1, TRIM28, TIF1β, HIV-1 gene expression, transcription, latency

## Abstract

HIV-1 latency maintenance and reactivation are regulated by several viral and host factors. One such factor is Krüppel-associated box (KRAB)-associated protein 1 (KAP1: also named TRIM28 or TIF1β). While initial studies have revealed KAP1 to be a positive regulator of latency reversal in transformed and primary CD4^+^ T cells, subsequent studies have proposed KAP1 to be a repressor required for latency maintenance. Given this discrepancy, in this study, we re-examine KAP1 transcription regulatory functions using a chemical genetics strategy to acutely deplete KAP1 expression to avoid the accumulation of indirect effects. Notably, KAP1 acute loss partially decreased HIV-1 promoter activity in response to activating signals, a function that can be restored upon complementation with exogenous KAP1, thus revealing that KAP1-mediated activation is on target. By combining comprehensive KAP1 domain deletion and mutagenesis in a cell-based reporter assay, we genetically defined the RING finger domain and an Intrinsically Disordered Region as key activating features. Together, our study solidifies the notion that KAP1 activates HIV-1 transcription by exploiting its multi-domain protein arrangement via previously unknown domains and functions.

## 1. Introduction

HIV-1 establishes persistent latent infections in the presence of suppressive antiretroviral therapy (ART) in several cell types including resting memory CD4^+^ T lymphocytes [1,2,3], monocytes and macrophages [4,5], and microglia in the brain [6], thus forming the so-called latent reservoir. These cells do not apparently produce viral products, thereby remaining refractory to immune surveillance mechanisms and complicating efforts to achieve a functional cure.

Although the molecular rules governing post-integration latency appear to be pleiotropic [7,8], one common feature is the resting state of the latently infected cell leading to low, or even undetectable, levels of viral transcription depending on the cell [9]. Despite several landmarks in the field, we currently lack a complete understanding of the fundamental regulatory principles of the HIV-1 transcriptional program and its implications for proviral fate control (latency maintenance and reactivation). Consequently, it is of great importance to uncouple molecular mechanisms that act directly on the provirus or indirectly via cell state alterations, thus enabling the identification of bona fide host factors for alternative strategies to sensitize latently infected cells for their elimination over bystander cells [10,11].

The HIV-1 transcriptional program is regulated at several levels: during cell homeostasis (basal phase), early in response to environmental stimulation (host phase), and by the subsequent activation by the viral encoded factor Tat (viral phase) [8,12]. The synchronized activation of the host and viral phases together promotes a positive feedback program for sustained viral transcription and replication, which are required to perpetuate the infection.

A large body of work has identified several viral and host factors that modulate the various phases of the HIV-1 transcriptional program, either positively (activation) or negatively (repression) [7,8,12]. In the host phase, when cells are exposed to immune stimulation, host transcription factors (TFs) like NF-κB and NFAT translocate from the cytoplasm into the nucleus to bind the proviral promoter leading to transcription activation [13,14]. These master TFs require co-regulators including members of the pre-initiation complex, chromatin-modifying enzymes, and elongation factors, among others [15,16]. The initial transcription activity during the host phase leads to Tat synthesis, which then takes over by creating the positive feedback loop that amplifies viral transcription by several orders of magnitude, by primarily operating at the elongation step [17,18].

During the last decade, one host factor that received increasing attention in the HIV-1 transcription field is KAP1, also known as TRIM28 and TIF1β [19]. KAP1 is a member of the tripartite motif (TRIM) family [20] and a multi-domain-containing protein involved in many functions. These include DNA repair [21], the reinstatement of heterochromatin after DNA replication [22], the transcription repression of endogenous retroviruses in embryonic stems cells and genes and retroelements in progenitor cells [23,24,25], and transcription activation [26,27]. KAP1 regulates critically important phenotypes such as cell growth, immune responses, differentiation, and development [21,28]. Given its importance to a multitude of biological processes, KAP1 has been linked to several pathologies including obesity [29], cancer [30], and the control of several pathogens (Reviewed in [31]).

In the context of HIV-1, KAP1 was initially linked to viral integration [32] and 5 years later to the control of viral transcription [26]. While two studies have linked KAP1 to viral transcription activation in transformed and primary CD4^+^ T cells [26,27], two other reports have associated KAP1 with transcription repression [33,34], perhaps attempting to correlate KAP1 to its historical role in repressing genes and retroelements in mouse embryonic stem cells and progenitor cells [35,36,37]. These conflicting data fueled the need in the field to revisit the experimental evidence to assess whether KAP1 activates and/or represses HIV-1 gene expression in different contexts [31].

To settle this controversy, we utilized an advanced genetic system allowing the CRISPR-mediated integration of the degron FKBP12^F36V^ in the C-terminus of endogenous KAP1 in HCT116 cells to allow for its acute depletion. In response to a bumped heterobifunctional dTAG molecule that induces dimerization of FKBP12^F36V^ fusion chimeras, the CRBN E3 ligase complex directs KAP1 to the proteasome for rapid protein degradation [38]. This approach allows testing for primary phenotypes to better delineate causality in transcriptional control [39], thus bypassing potential secondary effects of chronic factor silencing or depletion that could confound data interpretations. Using this system, we provide compelling evidence that cross-validates previous work demonstrating KAP1 as an activator of HIV-1 and not a repressor. Notably, by exploiting detailed deletion mapping and site-directed mutagenesis, we defined KAP1 domains and functions involved in HIV-1 promoter activation including its RING E3 ubiquitin ligase domain, its intrinsically disordered region (IDR), and nuclear localization, among others. Collectively, our efforts demonstrate that KAP1 activates the HIV-1 provirus using a combination of previously unknown functions, thus revealing the plasticity of a host factor to regulate viral gene expression.

## 2. Methods

### 2.1. Cell Culture

HEK293T (ATCC, Manassas, VA, USA, CRL-3216), HeLa (ATCC, Manassas, VA, USA, CCL-2), HCT116 (ATCC, Manassas, VA, USA, CCL-247), and HCT116:KAP1^dTAG^ cells were cultured in complete tissue culture media containing DMEM high glucose (Gibco, Thermo Fisher Scientific, Waltham, MA, USA, catalog 11-965-092), 7% fetal bovine serum (FBS), and 1% penicillin and streptomycin at 37 °C with 5% CO_2_. Mycoplasma testing was performed with a commercial kit (Southern Biotech, Birmingham, AL, USA, catalog 13100-01) to ensure that cells were not contaminated throughout the completion of this study.

### 2.2. Generation of the HCT116:KAP1^dTAG^ Cell Line

The HCT116:KAP1^dTAG^ cells were recently created [40]. Briefly, HCT116 parental cells were seeded in 6-well plates (~500,000 cells/well) and transfected ~24 h post-seeding with three plasmids (a blasticidin [BSD] arm plasmid, a puromycin [Puro] arm plasmid, and single guide [gRNA] plasmid) targeting the C-terminus of *KAP1*/*TRIM28* at a ratio of 1:1:1 (0.33 µg each). Before transfection, the media was changed into 1 mL of fresh, complete tissue culture DMEM media. Two mixes were prepared as follows: (1) a DNA mix consisting of 1 µg of total plasmid DNA (0.33 µg of each plasmid) into 100 µL of OPTIMEM (Gibco, Thermo Fisher Scientific, Waltham, MA, USA, catalog 11058021) and (2) a Lipofectamine 2000 (Invitrogen, Thermo Fisher Scientific, Waltham, MA, USA, catalog 52887) mix of 3 µL incubated in the final 100 µL of OPTIMEM. The Lipofectamine 2000 mix was added dropwise to the DNA mix and incubated for 10 min at room temperature. After incubation, the 200 µL mix was added dropwise to cells, and the tissue culture media was changed into 2 mL of complete tissue culture DMEM media after 5 h. Two days post transfection, selection media (1.5 µg/mL puromycin (Thermo Fisher Scientific, Waltham, MA, USA, catalog 227420500) and 10 µg/mL blasticidin (Thermo Fisher Scientific, Waltham, MA, USA, catalog BP264750) in complete DMEM tissue culture media) was added to the cells for 14 days to induce targeting of both copies of *KAP1*/*TRIM28*, at which point stable colonies began to expand. A single-selection approach including 0.5 µg of only the puromycin arm plasmid and 0.5 µg of the gRNA plasmid was included as control. Cells were re-split in a 6-well plate, grown for 4 days, and then expanded to a 10 cm tissue culture dish. Population lysates were collected and subjected to KAP1 Western blot analysis to verify targeting efficiency. Cells were single-cell sorted in 96-well plates using the UTSW Children’s Research Institute Moody Foundation Flow Cytometry Core in complete tissue culture DMEM media (200 µL per well with no antibiotic). Cells grew in single-cell format for ~3–5 weeks prior to slowly expanding to tissue culture dishes with increasing surface availability (24-well, 6-well, and then to 10 cm tissue culture dishes). To verify the accuracy of the targeting approach, the expression of tagged KAP1 was probed. As expected, the presence of the lower band corresponding to endogenous KAP1 completely disappeared.

### 2.3. Generation of the HCT116:KAP1^dTAG^ Cell Lines Stably Expressing GFP and KAP1

To generate lentivirus, HEK293T plated in a 6-well plate were transfected with 0.5 μg pTRIP-KAP1:FLAG or pTRIP-GFP vector, 0.5 μg psPAX2 (Addgene, Watertown, MA, USA, catalog 12260), and 0.1 μg pMD2.G (Addgene, Watertown, MA, USA, catalog 12259) using 3 μL of Polyjet (SignaGen, Frederick, MD, USA, catalog SL100688) per well. Cell supernatants were collected and cleared of cell debris via centrifugation at 300× *g* for 5 min, 48 h post transfection. Viral transduction was performed via spinoculation using 5 × 10^5^ HCT116:KAP1^dTAG^ cells, 10 μg/mL polybrene (MilliporeSigma, Burlington, MA, USA, catalog H9268), and unsupplemented tissue culture DMEM media to a final volume of 1 mL per well of a 6-well plate, and then spun down at room temperature for 1.5 h at 2900 rpm followed by 4 h incubation at 37 °C. Cells were spun down, the virus was removed, and cells were incubated in 2 mL of DMEM/7% FBS/1% Pen/Strep for 24 h. Cells were transferred to a 10 cm tissue culture dish for outgrowth, and 4 days later they were sorted by FACS (UTSW Children’s Research Institute Moody Foundation Flow Cytometry Core) by detecting red fluorescent protein (RFP) expression in RFP low- and high-expressing populations. KAP1 and GFP protein expression levels were determined using Western blot and flow cytometry, and the populations of cells expressing transgenes close to endogenous levels were selected for downstream experiments.

### 2.4. Cloning of KAP1 Deletion Constructs

Full-length KAP1 and deletion constructs were cloned into the pcDNA/4TO-3xFLAG vector using the 5′-HindIII and 3′-XhoI restriction sites. Deletion construct inserts were PCR amplified by designing custom-made primers (Appendix A) targeting specific regions of wild-type *KAP1/TRIM28* that had been previously cloned into pcDNA4/TO [26]. Amplified inserts and pcDNA4/TO were then digested for 3 h with HindIII-HF (NEB, Ipswich, MA, USA, catalog R3104) and XhoI (NEB, Ipswich, MA, USA, catalog R0146) using CutSmart Buffer. Ligated DNA was then transformed into *E. coli* DH5α cells (NEB, Ipswich, MA, USA, catalog 2987I) and grown overnight on LB + Ampicillin agar selection plates. Individual colonies were selected and grown in LB+ Ampicillin liquid media cultures. Plasmids were purified using the Plasmid MiniPrep Kit (Qiagen, Germantown, MD, USA, catalog 27106). Purified plasmids were verified to contain the correct inserts using HindIII and XhoI restriction digestion and Sanger sequencing.

### 2.5. Site-Directed Mutagenesis

KAP1 point mutants (Appendix A) were generated using QuikChange II XL Site-Directed Mutagenesis kit (Agilent, Santa Clara, CA, USA, catalog 200522) per manufacturer’s instructions. PCR-amplified DNAs were transformed into *E. coli* BME treated XL-10 Gold provided in the kit and positive clones validated by Sanger Sequencing.

### 2.6. Luciferase Reporter Assays

In Figure 1, HCT116:KAP1^dTAG^ cells were seeded into 24-well plates and transfected with a mix of DNAs (250 ng total DNA/well) and 0.75 µL of Polyjet (SignaGen, Frederick, MD, USA, catalog SL100688) per well as per the manufacturer’s instructions. Cells were transfected with pcDNA3.1-HIV-1–LTR-FFL LUC reporter vector (100 ng/well) [41], pRL-TK (Promega, Madison, WI, USA, catalog E2241) (25 ng/well), and carrier DNA (pBluescript II KS+) to complete 250 ng total DNA per well. Five h post transfection, the culture media was replaced with complete tissue culture media. After 16 h, cells were treated with 500 nM dTAG-13 (Tocris, Minneapolis, MN, USA, catalog 6605) or DMSO for 24 h followed by 4 h treatment with Human Tumor Necrosis Factor alpha (TNF-α) (PeproTech, Cranbury, NJ, USA, catalog 300-01A) or Phorbol 12-myristate 13-acetate (PMA) (MilliporeSigma, Burlington, MA, USA, catalog P8139). The samples were then collected for measurement on FLUOstar OPTIMA (BMG LABTECH, Ortenberg, Germany). For Figure 1a, FFL luciferase reporter activities were normalized to a constitutive pRL-TK using the Dual-Luciferase Reporter Assay System (Promega, Madison, WI, USA, catalog E1980). In Figure 2b,d and Figure 3b,d, HCT116:KAP1^dTAG^ cells were seeded into 24-well plates and transfected with a mix of DNAs (250 ng total DNA/well) and 0.75 µL of Polyjet per well. Cells were transfected with pcDNA3.1-HIV-1–LTR-FFL LUC reporter vector (50 ng/well), pRL-Null (Promega, catalog E2271) (5 ng/well), pcDNA4/TO GFP:SF (50 ng/well), pcDNA4/TO KAP1:F (50 ng/well) and/or pcDNA4/TO KAP1 deletion constructs or point mutants (50–100 ng/well), and carrier DNA (pBluescript II KS+) to complete 250 ng total DNA per well. Five h post transfection, the tissue culture media was replaced with fresh media. Then, samples were collected for measurement and FFL luciferase reporter activities were normalized to a constitutive pRL-Null using the Dual-Luciferase Reporter Assay System. In Figure 4a, HCT116:KAP1^dTAG^ cells were seeded into 24-well plates and transfected with a mix of DNAs (250 ng total DNA/well) and 0.75 μL of Polyjet (SignaGen, Frederick, MD, USA, catalog SL100688) per well as per the manufacturer’s instructions. Cells were transfected with pcDNA3.1-HIV-1–LTR-FFL LUC reporter vector (50 ng/well), pGL3-3xAP1-FFL (50 ng/well), pGL3-NF-κB-FFL (50 ng/well) [42] and/or pGL3-NFAT-FFL (100 ng/well) (Addgene, Watertown, MA, USA, catalog 17870), pRL-Null (5 ng/well), pcDNA4/TO GFP:SF (50 ng/well), pcDNA4/TO KAP1:F (50 ng/well) and/or pcDNA4/TO KAP1 RING Mut:F (75 ng/well), and carrier DNA (pBluescript II KS+) to complete 250 ng total DNA per well. Firefly luciferase reporter activities in Figure 4 were not normalized due to high levels KAP1 activation of the RL reporter, which blunted overall FFL signal without altering the phenotype. In Appendix A, HCT116:KAP1^dTAG^ cells were seeded into 24-well plates and transfected with a mix of DNAs (250 ng total DNA/well) and 0.75 µL of Polyjet per well. Cells were transfected with pcDNA3.1-HIV-1–LTR-FFL LUC reporter vector (50 ng/well), pRL-Null (Promega, catalog E2271) (5 ng/well), and carrier DNA (pBluescript II KS+) to complete 250 ng total DNA per well. Appendix A were completed using the same methods described for Figure 2 and Figure 3 in the indicated cell lines and conditions. For the reporter experiments conducted in the presence of ART (Appendix A), cells were treated for 16 h with an ART cocktail containing 100 nM Efavirenz (HIV Reagent Program, Manassas, VA, USA, HRP-4624), 200 nM Raltegravir (HIV Reagent Program, Manassas, VA, USA, HRP-11680), and 200 nM Saquinavir (HIV Reagent Program, Manassas, VA, USA, ARP-4658) or with vehicle DMSO.

### 2.7. Immunofluorescence and Confocal Microscopy

One day prior to seeding cells, coverslips were coated with poly-L-lysine (1:10 dilution) (MilliporeSigma, Burlington, MA, USA, catalog P8920) at room temperature and protected from light. HCT116:KAP1^dTAG^ cells were seeded onto coverslips in 24-well plates and transfected with a mix of DNAs (250 ng total DNA/well) and 0.75 µL of Polyjet per well. Cells were transfected with pcDNA/4TO plasmids expressing FLAG-tagged KAP1 or domains: KAP1:F (50 ng), KAP1ΔRING:F (100 ng), KAP1ΔIDR:F (50 ng), KAP1ΔIDR +NLS:F (50 ng), KAP1 NLS full mutant:F (50 ng), or KAP1 NLS full mutant +NLS (50 ng), and carrier DNA (pBluescript II KS+) to complete 250 ng total DNA per well. Five h post transfection, the tissue culture media was replaced and 24 h later, cells were fixed with 1% paraformaldehyde (PFA) (MilliporeSigma, Burlington, MA, USA, catalog I58127) for 5 min at room temperature. After fixation, cells were washed three times with 1X PBS. Samples were permeabilized with 0.5% Triton X-100 for 2 min at 4 °C followed by three washes with 1X PBS. Blocking was performed using IF blocking solution containing 5% normal goat serum (NGS) (Abcam, Waltham, MA, USA, catalog ab7481) and 4% bovine serum albumin (BSA) (RPI, Mount Prospect, IL, USA, catalog A30075) in 1X PBS for 1 h followed by 1 h incubation in mouse anti-FLAG M2 primary antibody (1:10,000) at room temperature (Appendix A). Coverslips were washed three times with 1X PBS and then incubated in goat anti-mouse AF488-conjugated secondary antibody (1:500) at room temperature (Appendix A), followed by three washes with 1X PBS and then dried at room temperature. Coverslips were then mounted onto pre-cleaned fluorescent antibody glass slides (Thermo Fisher Scientific, Waltham, MA, USA, catalog 3032-002) using Vectashield Antifade Mounting Medium with DAPI (Vector Laboratories, Newark, CA, USA, catalog H-1200) and cured for 24 h protected from light. Samples were imaged using an LSM510 confocal microscope (Zeiss, Dublin, CA, USA) with x63 oil immersion objective at the UTSW Quantitative Light Microscopy Core Facility.

### 2.8. Western Blots

Total protein extracts of the indicated cell lines and conditions were subjected to SDS-PAGE followed by Western blot. Nitrocellulose membranes were blocked in 5% non-fat dry milk and TBS-T (Tris-buffered saline containing 0.2% Tween-20) for 1 h and incubated with primary antibodies at 4 °C from 1 h to overnight (Appendix A). Membranes were then exposed to Clarity Western ECL substrate and imaged using a ChemiDoc (Bio-Rad, Hercules, CA, USA).

## 3. Results

### 3.1. Acute KAP1 Depletion Partially Decreases Signal-Induced HIV-1 Reporter Activation

Previous contradictory studies proposed that KAP1 both activates and represses HIV-1 gene expression via chronically silenced KAP1 in the same cell models of latency. To settle this controversy, we leveraged an advanced genetic (dTAG) system for rapid protein degradation [38]. The dTAG approach involves genome editing in a diploid cell line to tag endogenous *KAP1/TRIM28* with an FKBP12^F36V^ degron, which mediates protein degradation following the addition of the dTAG-13 ligand. This degron is followed by a 2x-HA tag, a P2A protease cleavage site, and BSD or Puro cassettes for the selection of resistance cells (Figure 1a). Given the issues with pseudo ploidy in Jurkat [43] and multiple *KAP1* copies making the dTAG approach cumbersome, we chose the HCT116 diploid cell line that is commonly used for mechanistic studies. We created an HCT116 clone (here referred to as HCT116:KAP1^dTAG^) with biallelic knock-in of the FKBP12^F36V^-HA tag at the KAP1 carboxy terminus (Figure 1a). Treating the HCT116:KAP1^dTAG^ cell line with dTAG-13 (0.5 µM) for 4, 8, and 24 h yielded a ~50%, ~90%, and nearly complete decrease in KAP1 expression, respectively, relative to DMSO treatment for 24 h (Figure 1b), thus confirming the efficiency of KAP1 acute depletion.

**Figure 1 viruses-16-00116-f001:**
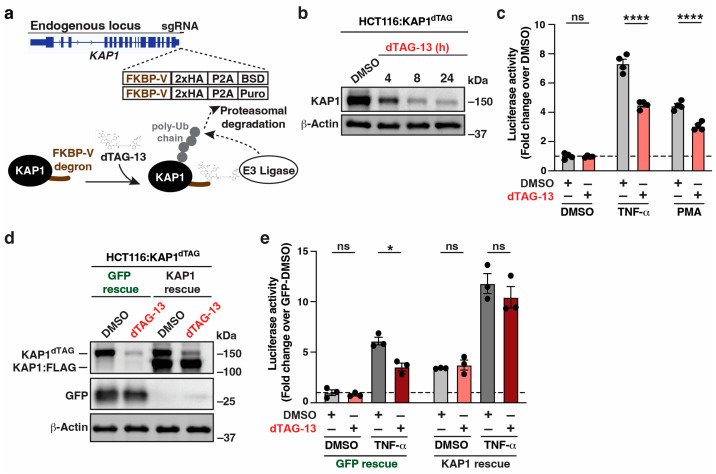
Acute KAP1 depletion partially decreases signal-induced HIV-1 reporter activation. (**a**) Top, scheme of *KAP1*/*TRIM28* endogenous locus engineered through CRISPR-In in the 3′-end in the HCT116:KAP1^dTAG^ cell line. Bottom, scheme showing dTAG-13–mediated targeting of the CRBN E3 ubiquitin ligase to KAP1 tagged with a degron sequence and its proteasome-mediated degradation. (**b**) Western blot analysis with the indicated antibodies showing KAP1 depletion upon dTAG-13 treatment in the HCT116:KAP1^dTAG^ cell line. (**c**) Luciferase assay in the HCT116:KAP1^dTAG^ cell line transiently transfected with the HIV-1 FFL and pRL-TK luciferase reporters for 16 h, pre-treated with DMSO or dTAG-13 for 24 h and treated with the indicated ligands (Vehicle DMSO, TNF-α or PMA) for 4 h. Data represent mean +/– SEM. Fold luciferase activity (N = 4, two-way ANOVA followed by Sidak’s multiple comparison test). **** *p* < 0.0001, and ns = non-significant. (**d**) Western blot analysis with the indicated antibodies in the HCT116:KAP1^dTAG^ cell line reconstituted with GFP or FLAG-tagged KAP1 and treated with DMSO or dTAG-13. (**e**) Luciferase assay in the HCT116:KAP1^dTAG^ cell line reconstituted with GFP or FLAG-tagged KAP1 transiently transfected with the HIV-1 FFL and pRL-TK luciferase reporters for 16 h and treated with DMSO or dTAG-13 for 24 h. Data represent mean +/– SEM. Fold luciferase activity (N = 3, two-way ANOVA followed by Sidak’s multiple comparison test). * *p* < 0.05, and ns = non-significant.

To assess whether KAP1 acute loss impacts basal and/or signal-induced HIV-1 gene expression, we transfected the HCT116:KAP1^dTAG^ cell line with a vector containing the HIV-1 promoter driving Firefly (FFL) luciferase (HIV-1 reporter) [41] for 24 h, pre-treated cells with DMSO (vehicle control) or dTAG-13 for 24 h, and finally treated cells with either DMSO, TNF-α (pro-inflammatory cytokine), or PMA (PKC agonist) for 4 h to stimulate LTR-driven transcription. To account for variations in transfection efficiencies, the HIV-1 FFL luciferase activity was normalized to an internal Renilla-expressing vector (see Methods). Notably, while acute KAP1 depletion had no effect on basal (DMSO treated) HIV-1 reporter activity (Figure 1c), KAP1 loss revealed a statistically significant drop in both TNF-α– and PMA–induced HIV-1 reporter activity, consistent with previous studies linking KAP1 to signal-induced HIV-1 transcription activation [26,27] and to oncogenic signaling [30].

Importantly, to further evaluate whether decreased HIV-1 reporter activation is on target, we reconstituted the HCT116:KAP1^dTAG^ cell line with pTRIP lentiviruses ectopically expressing GFP (negative control) or KAP1 at nearly endogenous levels (Figure 1d). Exogenous KAP1, but not GFP, rescued TNF-α–mediated HIV-1 reporter activation (Figure 1e), thus providing further evidence that KAP1 can activate the HIV-1 promoter and rescue the dampened HIV-1 expression upon acute KAP1 depletion. Additionally, KAP1 ectopic expression increased basal HIV-1 reporter expression, partially bypassing the need for a ligand, which is discussed later (Figure 2, Figure 3 and Figure 4).

ART is used in clinics to block viral replication by targeting various stages in the HIV-1 life cycle without interfering with the transcription process. To ascertain if endogenous KAP1 can facilitate signal-induced activation of the HIV-1 promoter in the presence of ART, HCT116:KAP1^dTAG^ cells were transiently transfected with the HIV-1 reporter, KAP1 was depleted with dTAG-13, and the cells were then pre-treated with ART or vehicle (DMSO) and finally challenged with TNF-α or vehicle (PBS). As expected, the presence of ART in the tissue culture has no impact on signal-induced KAP1 activation of the HIV-1 promoter (Appendix A). Together, these data cross-validate prior results in immortalized and primary CD4^+^ T cells providing compelling evidence that KAP1 is required for the full activation of the HIV-1 promoter [26,27] and that KAP1 could function in reservoir cells irrespective of ART to promote low-level HIV-1 gene expression.

Thus far, we have established that endogenous KAP1 is required for full expression of the HIV-1 reporter upon signal induction and that KAP1 ectopic expression also induces HIV-1 reporter activity in HCT116 cells (Figure 1e). To test if this phenotype was cell-type specific, we co-transfected two additional cell lines (HEK293T and HeLa) with the HIV-1 reporter alongside FLAG-tagged KAP1 or GFP. Importantly, ectopic KAP1 expression increased the basal activity of the HIV-1 promoter in the three cell lines evaluated, albeit at different fold levels (Appendix A), signifying that the KAP1 activating function is cell-type independent and not restricted to HCT116. Additionally, TNF-α stimulation has an additive effect in all cell lines, which is in line with previous data indicating that ectopic KAP1 expression can partially overcome liganded (TNF-α)-driven stimulation of the HIV-1 promoter (Figure 1e). Finally, ART does not alter the activation of the HIV-1 promoter upon dual ectopic KAP1 expression and TNF-α stimulation (Appendix A), again suggesting that KAP1 could function in reservoir cells irrespective of ART to promote low-level HIV-1 gene expression.

### 3.2. KAP1 Deletion Analysis Predicts Domains Important for HIV-1 Reporter Activation

Despite validating KAP1’s ability to act as a transcription activator using the acute depletion system (HCT116:KAP1^dTAG^), the mechanism of activation remains unclear. KAP1 is a multi-domain protein including the N-terminal, RING finger (R), B1 and B2 boxes (B), coiled coil (CC), and the intrinsically disordered region (IDR) connecting the RBCC and the tandem C-terminal Plant-homology domain (PHD) and Bromo domain (BD) cassette (Figure 2a). Some of these domains were characterized biochemically and/or genetically, including the RING finger functioning as an E3 ubiquitin ligase [44], the B1 box and CC known to promote KAP1 oligomerization and homo-dimerization [45,46], respectively, and the PHD domain involved in the intra SUMOylation of the BD [47] and chromatin binding [30].

**Figure 2 viruses-16-00116-f002:**
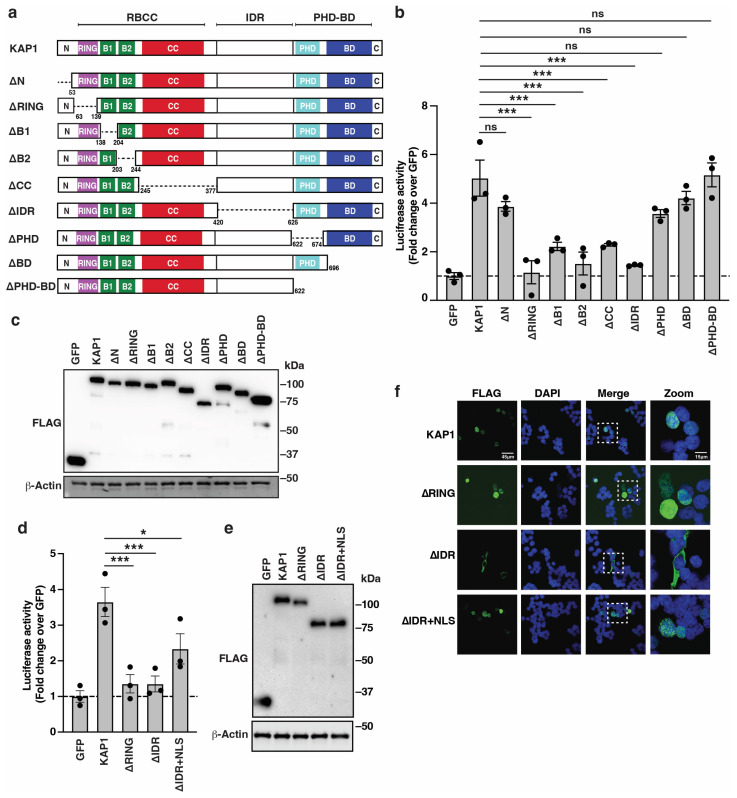
KAP1 deletion analysis predicts domains important for HIV-1 reporter activation. (**a**) Schematic of KAP1 and domain deletions. (**b**) Luciferase assay in the HCT116:KAP1^dTAG^ cell line transiently transfected with the HIV-1 FFL and pRL-Null luciferase reporters for 24 h alongside FLAG-tagged GFP or KAP1 (full-length and deletion constructs). Data represent mean +/– SEM. Fold luciferase activity (N = 3, one-way ANOVA followed by Dunnett’s test for multiple comparisons to KAP1). *** *p* < 0.001, and ns = non-significant. (**c**) Western blot analysis of HCT116:KAP1^dTAG^ cells co-transfected with HIV-1 FFL, pRL Null, and FLAG-tagged GFP or KAP1 (full-length or deletion constructs). Blots are representative of three independent experiments. (**d**) Luciferase assay in the HCT116:KAP1^dTAG^ cell line transiently transfected with the HIV-1 FFL and pRL-Null luciferase reporters for 24 h alongside FLAG-tagged GFP or KAP1 (full-length and deletion constructs). Data represent mean +/– SEM. Fold luciferase activity (N = 3, one-way ANOVA followed by Dunnett’s test for multiple comparisons to KAP1). * *p* < 0.05, *** *p* < 0.001, and ns = non-significant. (**e**) Western blot analysis of HCT116:KAP1^dTAG^ cells co-transfected with HIV-1 FFL, pRL-Null, and FLAG-tagged GFP or KAP1 (full-length or deletion constructs). Blots are representative of three independent experiments. (**f**) Confocal images of indirect immunofluorescence of ectopically expressed FLAG-tagged constructs in HCT116:KAP1^dTAG^ cells.

To define KAP1 domains and activities required for HIV-1 promoter activation, we co-transfected HCT116:KAP1^dTAG^ cells with the HIV-1 reporter alongside FLAG-tagged full-length KAP1 and constructs bearing the deletions of the individual KAP1 domains (Figure 2a) or GFP as baseline control. Since ectopic KAP1 expression equally increased basal HIV-1 reporter activity in HCT116:KAP1^dTAG^ cells treated or not with dTAG-13 (Appendix A) and partially bypassed the need for the ligand (TNF-α) (Figure 1e and Appendix A), perhaps due to KAP1 over-expression, the cells were just transfected with the various constructs in the absence of dTAG-13 treatment. To account for variations in transfection efficiencies, the HIV-1 FFL luciferase activity was normalized to an internal Renilla-expressing vector (see Methods). In this assay, five domains (RING, B1, B2, CC, and IDR) were required for KAP1 activation because their deletion consistently yielded statistically significant lower HIV-1 reporter activities relative to cells transfected with full-length KAP1 (Figure 2b). Conversely, the deletion of three domains (N, PHD, and BD) did not have strong functional consequences, suggesting that they are not required in this context. Importantly, all KAP1 deletion constructs expressed at similar, if not identical, levels (Figure 2c), signifying that domain deletion does not largely affect KAP1 protein expression.

To test if the phenotypes of the various KAP1 deletions were masked from the conceivable multimerization of the ectopically expressed constructs with endogenous KAP1, the experiment was repeated in the presence of dTAG-13 or vehicle DMSO to virtually eliminate this possibility. Notably, the KAP1 deletions equally activated the HIV-1 promoter in both contexts (Appendix A), thus ruling out the possibility of multimerization between endogenous KAP1 and the ectopically expressed KAP1 constructs.

The loss of KAP1 activation upon IDR deletion (Figure 2b) was consistent with the presence of a predicted NLS within the IDR [48], which impedes a critical assessment of HIV-1 reporter activation by the IDR if the protein loses nuclear localization. To investigate this idea further, the SV40 T-Ag NLS (PPKKKRKV) was appended to the C-terminus of KAP1ΔIDR (KAP1ΔIDR + NLS) to compare its HIV-1 reporter activity to full-length KAP1, KAP1ΔIDR without the appended NLS, and the construct with the largest trans-activation defect (KAP1ΔRING).

While IDR deletion (KAP1ΔIDR) virtually dampened HIV-1 reporter activity, the addition of the SV40 T-Ag NLS (KAP1ΔIDR + NLS) partially (~50%) restored HIV-1 reporter activity (Figure 2d), and both proteins expressed similarly (Figure 2e), suggesting that the IDR has previously unknown functions unrelated to maintaining the proper nuclear localization of KAP1. To further validate if the reduced HIV-1 reporter activation by KAP1ΔIDR and KAP1ΔRING occurs due to the loss of nuclear localization, we performed FLAG immunofluorescence followed by confocal microscopy analysis. As expected, KAP1ΔIDR localized to the cytoplasm, and its correct nuclear localization was restored upon the addition of the heterologous NLS (Figure 2f). Notably, both KAP1 and KAP1ΔRING localized to the nucleus (Figure 2f), suggesting that the function of the RING finger domain is unrelated to the regulation of nuclear localization.

Taken together, the deletion mapping analysis followed by expression and subcellular localization analysis revealed protein domains and enzymatic activities (E3 ligase) potentially required for HIV-1 promoter activation. Additionally, the deletion of the IDR decreased HIV-1 reporter activity because of a loss in nuclear localization, which can be partially restored by appending a heterologous NLS.

### 3.3. KAP1 RING Finger, Nuclear Localization, and IDR Are Critical Determinants for HIV-1 Reporter Activation

The deletion mapping analysis revealed KAP1 domains necessary for HIV-1 promoter activation in cells (Figure 2). However, since large deletions could alter protein folding and function, we cross-validated these data using precision mutations known to biochemically perturb specific functions including the RING (C65A) [44], B1 box (A160D/T163A/E175R) [45,46], CC (V293S/K296A/M297A/L300S) [45,46], HP1 box within the IDR (V488E), and predicted NLS [48] (Figure 3a). Since the NLS is not fully characterized [48], we created NLS partial mutant 1 [PM1] (K469A/R470A), NLS partial mutant 2 [PM2] (R483A/K484A), and NLS full mutant (K469A/R470A/R483A/K484A).

**Figure 3 viruses-16-00116-f003:**
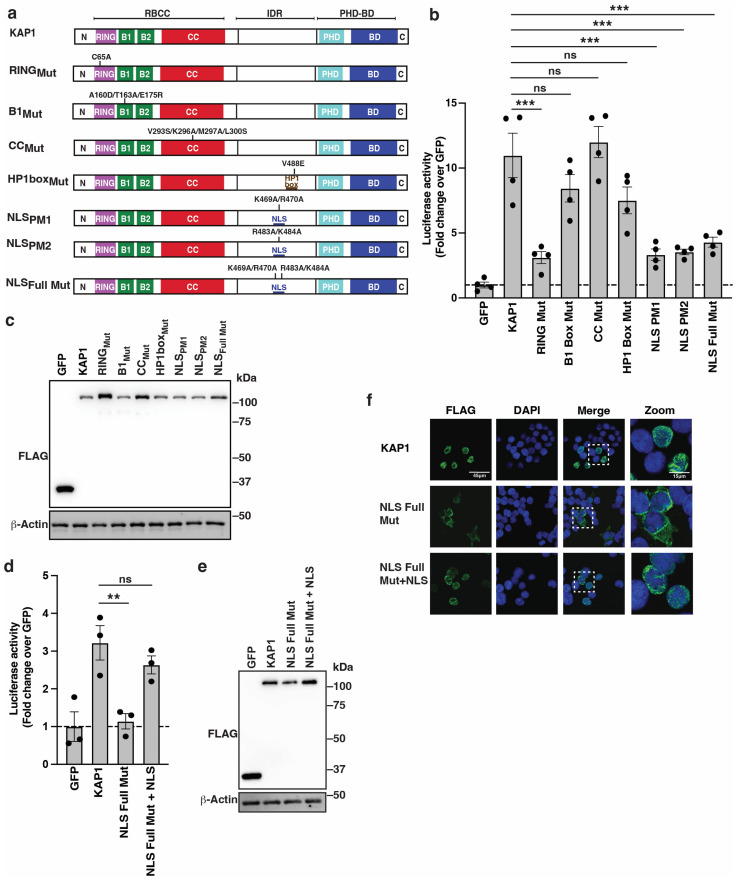
KAP1 nuclear localization and RING E3 ubiquitin ligase function are required for HIV-1 reporter activation. (**a**) Schematic of KAP1 and point mutations. (**b**) Luciferase assay in the HCT116:KAP1^dTAG^ cell line transiently transfected with the HIV-1 FFL and pRL-Null luciferase reporters for 24 h alongside FLAG-tagged GFP or KAP1 (full-length and deletion constructs). Data represent mean +/– SEM. Fold luciferase activity (N = 4, one-way ANOVA followed by Dunnett’s test for multiple comparisons to KAP1). *** *p* < 0.001, and ns = non-significant. (**c**) Western blot analysis of HCT116:KAP1^dTAG^ cells co-transfected with HIV-1 FFL, pRL Null, and FLAG-tagged GFP or KAP1 (full-length or point mutants). Blots are representative of four independent experiments. (**d**) Luciferase assay in the HCT116:KAP1^dTAG^ cell line transiently transfected with the HIV-1 FFL and pRL-Null luciferase reporters for 24 h alongside FLAG-tagged GFP or KAP1 (full-length and deletion constructs). Data represent mean +/– SEM. Fold luciferase activity (N = 3, one-way ANOVA followed by Dunnett’s test for multiple comparisons to KAP1). ** *p* < 0.01, and ns = non-significant. (**e**) Western blot analysis of HCT116:KAP1^dTAG^ cells co-transfected with HIV-1 FFL, pRL Null, and FLAG-tagged GFP or KAP1 (wild-type and indicated constructs). Blots are representative of three independent experiments. (**f**) Confocal images of indirect immunofluorescence of ectopically expressed FLAG-tagged constructs in HCT116:KAP1^dTAG^ cells.

FLAG-tagged KAP1 (full-length and point mutants) or GFP as baseline control were co-transfected into HCT116:KAP1^dTAG^ cells alongside the HIV-1 FFL reporter and pRL-Null for normalization purposes, and luciferase activity was recorded 24 h post transfection. The mutation data suggest three classes of phenotypes: (1) mutations that virtually ablate KAP1 activation (RING finger and NLS mutants), (2) mutations that partially dampen KAP1 activity (B1 box and HP1 box mutant) albeit not statistically significant, and (3) mutations conferring no effect (CC mutant) (Figure 3b). Importantly, these mutants express at similar, if not identical, levels relative to wild-type KAP1 (Figure 3c), thus ruling out compromised KAP1 mutant activity because of reduced protein levels.

The B1 box and CC point mutants did not decrease much activity compared to full deletion of the B1 and CC domains (Figure 2b), suggesting that high-order KAP1 organization (oligomerization and asymmetric dimerization, respectively) may not be necessary for KAP1 activation. Similarly, the mutation of the HP1 box within the IDR had a small effect on KAP1 activation relative to the deletion of the entire IDR (Figure 2b), perhaps due to the IDR containing additional domains for KAP1 activation including the NLS. As expected, the mutation of the RING and NLS (one or both motifs) almost completely (~3–4-fold) diminished KAP1 activity (Figure 3b), potentially indicating the need for a previously unknown KAP1 E3 ligase function in the nucleus for full HIV-1 promoter activation.

To cross-validate the importance of KAP1 nuclear localization to HIV-1 reporter activation, the SV40 T-Ag NLS was appended to the C-terminal of the KAP1 full NLS mutant and its activity compared to full-length KAP1 and to the virtually inactive KAP1 full NLS mutant. Notably, the addition of the heterologous NLS to a non-functional KAP1 NLS mutant almost completely restored KAP1 activity relative to KAP1 full-length (Figure 3d), and all constructs expressed at similar levels (Figure 3e). Consistent with this data, while the KAP1 NLS full mutant is cytoplasmic, the addition of the heterologous NLS restored its nuclear localization (Figure 3f). Collectively, we have validated the requirement of the RING finger E3 ligase domain and nuclear localization for full HIV-1 promoter activation.

Because ectopic KAP1 expression activated the HIV-1 promoter alone and in an additive manner with a stimulating ligand (TNF-α), we tested whether the non-functional mutants (ΔRING, ΔIDR, RING mutant and NLS full mutant) negatively impacted TNF-α stimulation. However, the expression of the non-functional mutants did not largely alter the activation of the HIV-1 promoter in response to TNF-α (Appendix A). Additionally, while ectopic KAP1 activated the HIV-1 promoter alone and in an additive manner with TNF-α stimulation in the three cell lines tested (HCT116:KAP1^dTAG^, HEK293T and HeLa), the expression of the non-functional RING mutant did not compromise activation of the HIV-1 promoter upon TNF-α stimulation in any cell line (Appendix A). These data are consistent with the idea that the KAP1 deletion constructs do not work as dominant negative inhibitors by multimerizing with and altering the activity of endogenous KAP1 (Appendix A).

### 3.4. The RING E3 Ligase Is Potentially Required in the Nucleus for a General, but Not Pathway-Specific, Transcription Activation Function

The above data suggested a requirement of the RING finger domain bearing the E3 ligase in the nucleus for HIV-1 promoter activation. If the KAP1 E3 ligase function is required to activate a specific cell signaling-transcriptional program in the cytoplasm or nucleus, we should observe pathway selectivity. The HIV-1 5′-LTR has dozens of transcription binding sites including the signal-responsive *cis* motifs for AP-1, NF-κB, and NFAT, among others [8]. Given our data illustrating a signal-induction requirement for HIV-1 activation by KAP1 (this study, and [26,27]), we used FFL luciferase reporters containing a minimal TATA-box promoter for pre-initiation complex activation through TATA-box binding protein (TBP) binding and three consecutive *cis* elements for each of the signal-responsive TFs (AP-1, NF-κB, and NFAT) in reporter assays in HCT116:KAP1^dTAG^ cells co-transfected with GFP, KAP1, or KAP1 RING mutant. Notably, the data indicate that KAP1, but not the KAP1 RING mutant, activates all three reporters in addition to the HIV-1 reporter, albeit at different levels (Figure 4a), suggesting a lack of promoter specificity.

Based on the collective evidence, KAP1 activation through the nuclear RING E3 ligase function extends beyond HIV-1 to promoters containing similar *cis* element arrangements, consistent with signal-induced transcription activation (Figure 1). Collectively, KAP1 may ubiquitinate one or more substrates, potentially general transcription factors (GTF), and/or other regulatory factors required for the global, but not pathway-specific activation of signal-induced transcriptional programs (Figure 4b).

**Figure 4 viruses-16-00116-f004:**
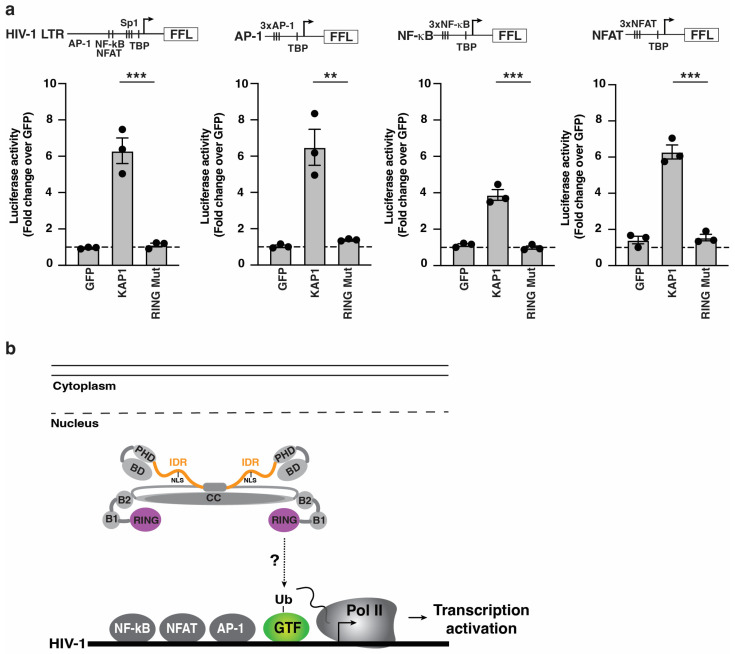
The KAP1 RING finger domain is required for activation of a variety of *cis*-element–driven promoters in the cell nucleus. (**a**) Luciferase assay in the HCT116:KAP1^dTAG^ cell line transiently transfected with the indicated FFL luciferase reporters (HIV-1, AP-1, NF-κB, and NFAT) for 24 h alongside FLAG-tagged GFP or KAP1 (wild-type and RING mutant). Data represent mean +/– SEM. Fold luciferase activity (N = 3, one-way ANOVA followed by Dunnett’s test for multiple comparisons to KAP1). ** *p* < 0.01, and *** *p* < 0.001. (**b**) Simple scheme describing the importance of KAP1 RING finger domain in the nucleus for HIV-1 transcription activation.

## 4. Discussion

Using an acute factor depletion system to mitigate issues with long-term silencing and depletion, we demonstrated KAP1′s ability to activate the HIV-1 promoter (Figure 1). While this approach cannot yet be implemented in primary CD4^+^ T cells, our data are in line with a large body of work in which KAP1 activates integrated proviruses in both transformed and primary CD4^+^ T cell models of latency [26,27], strongly indicating that KAP1 is required for signal-induced transcriptional activation and thus latency reactivation, but not latency maintenance [33,34]. Notably, KAP1 activates the HIV-1 promoter in response to potent stimulation cues (TNF-α and PMA) (as shown in this study and in [26,27]) and less potent latency-reversing agents (e.g., the histone deacetylase inhibitor Vorinostat, and the PKC agonist Bryostatin) [26,27], suggesting that KAP1 can facilitate HIV-1 transcription in response to a variety of ligands with a broad range of potencies. Our studies were not aimed at addressing how cell stimulation promotes KAP1 activation of the HIV-1 promoter. Nonetheless, we speculate that in response to stimulating signals, KAP1 may re-localize within the cell nuclei to assemble transcription-competent complexes at the provirus and that ectopic KAP1 can partially bypass the need of cell stimulation, perhaps due to the increased source of available KAP1 in the cell nuclei, artificially favoring molecular crowding events.

Our prior studies have shown the importance of KAP1 activation of the HIV-1 promoter in CD4^+^ T cells [26,27]. Because HIV-1 can infect other cell types in the periphery or tissue-resident state, we examined Human Protein Atlas datasets [49,50] and found that KAP1 is ubiquitously expressed in the immune system (Appendix A), potentially highlighting its importance in other non-lymphoid cell lineages, including myeloid. Given KAP1’s ubiquitous expression and the importance of reservoirs other than CD4^+^ T cells for viral persistence, including macrophages and microglia, future studies should investigate how KAP1 controls HIV-1 latency reactivation in other physiologically relevant reservoirs.

By exploiting an extensive deletion mapping and site-directed mutagenesis analysis, we revealed the genetic basis of KAP1 activation of the HIV-1 promoter. The deletion of five domains (RING, B1 box, B2 box, CC, and IDR), but not the C-terminal PHD and BD, significantly decreased reporter activity (Figure 2), suggesting KAP1 domain selectivity for HIV-1 promoter activation. To cross-validate the deletion mapping analysis and rule out artifacts due to large deletions possibly altering protein folding and activity, a site-directed mutagenesis approach was used to inactivate select domains (RING, B1, CC, and IDR) with precision point mutants inactivating specific functions (Figure 3).

Three classes of mutant phenotypes were identified. First, mutants in domains causing near-complete KAP1 inactivation including the RING finger domain and the predicted NLS within the IDR. Second, mutants in domains causing partial, albeit non-statistically significant, KAP1 inactivation including the B1 box involved in homo-oligomerization [46] and the HP1 box (PxVxL, short linear motif (SLiM)) [51] within the IDR required for binding to all three HP1 family members [52]. This lack of statistical significance can be attributed to the low dynamic range of the transcription reporter assay, and thus we cannot completely rule out the biological importance of the B1 and HP1 boxes. Third, mutants in domains causing no effects such as in the CC domain, which is required for asymmetric dimerization [46,53], perhaps signifying a lack of dimerization requirement for KAP1 activation in contrast to KAP1 silencing of retroelements. While deletion of the B2 box appears important for KAP1 activation of the HIV-1 promoter, the lack of known mutations linked to a particular function did not allow us to validate its importance [45,46], so its function in the context of transcriptional activation remains to be further elucidated.

A comparison of the deletion and mutagenesis datasets illustrates a lack of complete overlap. For example, while deletions of the B1 box and CC domain have large effects, their point mutations have little or no effect. This result potentially illustrates the deleterious effects of the deletions and/or previously unknown functions of residues other than those mutated to inactivate particular functions regulated by the B1 box and CC domain, such as oligomerization and asymmetric dimerization, respectively [46]. Similarly, deletion of the long IDR has large functional consequences, perhaps because it contains several SLiMs required for multiple non-overlapping functions including the HP1 box containing the PxVxL motif and the NLS. The HP1 box has been previously studied in the context of gene repression [54]. Thus, there is still the possibility that the KAP1 interaction with one or more HP1 family members may be cooperatively required for HIV-1 promoter activation. Testing these ideas will require rigorous genetic and biochemical approaches beyond the scope of this work.

The NLS is expectedly critical for KAP1 nuclear localization to facilitate HIV-1 promoter activation. While a previous study found that the deletion of residues 462–494 prevented KAP1 nuclear localization [48], the precise residues contributing to nuclear localization were not mapped. Notably, our site-directed mutagenesis approach accurately defined four non-contiguous amino acids within the IDR (K469 and R470 in motif 1 and R483 and K484 in motif 2) critical for KAP1 nuclear localization and subsequent HIV-1 promoter activation. The mutation of either motif virtually dampened KAP1 activation of the HIV-1 promoter without causing any additive effects, thus illustrating their independent requirement.

While the deletion of the IDR and mutation of the NLS within the IDR blocked nuclear localization, the deletion and point mutation of the RING finger domain did not alter KAP1’s nuclear localization, thus strongly indicating that the loss of HIV-1 promoter activation is independent of an abnormal change in subcellular localization. Because NLS mutation prevented both KAP1 nuclear localization and activation, we sought to uncouple these two effects by appending a heterologous NLS to the KAP1 C-terminal. While KAP1 NLS full mutant is expectedly cytoplasmic, the KAP1 NLS full mutant + NLS restored nuclear localization and full HIV-1 promoter activation, strongly indicating that nuclear localization is necessary for this activity. However, appending the heterologous NLS to KAP1ΔIDR only partially restored HIV-1 promoter activation despite the nuclear localization being retained, again illuminating a requirement for the IDR beyond nuclear localization, perhaps linked to the loss of the HP1 box and/or other functionally relevant SLiMs. Notably, the virtual loss of KAP1 activity in the RING finger E3 ligase domain mutant suggests a previously unknown involvement in HIV-1 promoter activation through substrate ubiquitination and/or SUMOylation. While previous proteomic screens have identified putative E3 ubiquitin ligase substrates [44,55,56], further validation and mechanistic investigations are required to test the involvement of these and/or other potential substrates in HIV-1 promoter activation. Surprisingly, Ma et al. reported the involvement of the KAP1 RING finger domain in CDK9 SUMOylation, but not ubiquitination, a modification that was required for HIV-1 transcriptional silencing [33]. While our studies and those by Ma et al. agree on the importance of the KAP1 RING domain, both associate it with different post-translational modifications and transcriptional outputs. Thus, future work is needed to first identify KAP1 ubiquitination and/or SUMOylation substrates to delineate their roles in the HIV-1 transcriptional cycle in the context of latency maintenance and reactivation. A potentially captivating substrate identified by Watanabe et al. in a proteomic screen is TFIIB [56], a GTF required for both transcription pre-initiation assembly at promoters and dissociation from the early Pol II transcribing complex [57,58,59], perhaps suggesting that KAP1-mediated TFIIB ubiquitination may tune two disparate steps (assembly and disassembly) of Pol II transcription complexes during transcription activation (Figure 4b).

Another interesting point for future research is the dual activating and repressive roles of KAP1 at different loci. It is noteworthy to mention that the protein domains and functions involved in KAP1-mediated gene repression are better understood. For example, KAP1 is recruited to the Primer Binding Site (PBS) of the integrated murine leukemia virus by a KRAB-domain-containing ZFP protein [60] through HP1 recruitment via the HP1 box [61] and SUMOylation of specific residues within the BD [62] to promote transcriptional silencing in embryonic stem and carcinoma cells. Additionally, KAP1 also represses the retrovirus PFV using its RBCC domain to recruit select KRAB-ZFPs family members [46,63]. Collectively, these studies illustrate the plasticity of KAP1 to perform two opposite functions (activation and repression) at different loci, perhaps using different protein domains and activities. However, the molecular and structural details of both these activities remain poorly understood so far.

Taken together, we have exploited advanced genetic systems to validate that KAP1 activates the HIV-1 promoter. This activation is not restricted to HIV-1 as minimal promoters bearing single signal-responsive *cis*-element motifs (AP-1, NF-κB, and NFAT) are indistinguishably activated, thus suggesting broad KAP1 activation of signal-inducible transcriptional programs. Future work will help identify KAP1 RING E3 ligase substrates and mechanistic details in the context of transcription, HIV-1 latency, and other physiologically and pathologically relevant contexts.

## Data Availability

Primary uncropped images and gel blots can be found online at MDPI.

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
