# Peer review of "Functional Analysis of KAP1/TRIM28 Requirements for HIV-1 Transcription Activation"

_viruses, 2024, doi:10.3390/v16010116_

Round 1

Reviewer 1 Report

Comments and Suggestions for Authors

The maintenance and reactivation of HIV-1 latency are under the control of various factors, which includes KAP1, the main player in the present study by Randolph et al. While KAP1 has been reported to act as a positive regulator in reversing latency, subsequent studies have suggested that KAP1 serves as a repressor necessary for maintaining latency. To address this inconsistency, the authors reevaluated the role of KAP1 in HIV-1 transcriptional regulation. To do so, they employed an HCT116 model cell line system in which KAP1 could be degraded rapidly at will, thus offsetting indirect effects that could accumulate in cells with prolonged depletion of KAP1. The authors showed that the acute loss of KAP1 reduced activity of the transfected HIV-1 reporter in response to activating signals, which could be restored by exogenous KAP1. They also found that the exogenous KAP1 increased basal levels of the HIV-1 reporter and a set of core promoters driven by specific transcription factors in the presence of endogenous KAP1. Importantly, through using a combination of comprehensive KAP1 deletion and point-mutation mutant proteins, the authors genetically identified four elements in the N-terminal half of KAP1 and a centrally located IDR as pivotal for HIV-1 reporter transactivation.

This is a well-written manuscript that shows convincingly the ability of KAP1 to promote HIV-1 transcription, suggesting a role of KAP1 in reversing and not maintaining a transcriptional form of HIV-1 latency. I only have one concern that should be addressed prior to acceptance of this work for publication.

Figure 2: The authors identified five regions in KAP1 important for HIV-1 reporter activity by using the HCT116:KAP1dTAG cell line. However, they did not take advantage of depleting the degron-tagged KAP1. Namely, there is a chance that the presence of the tagged KAP1 might have obscured phenotype of some of the deletion constructs, i.e. by multimerizing with them.   

Author Response

The responses are attached as PDF

Reviewer 2 Report

Comments and Suggestions for Authors

In this manuscript, Randolph et al have employed a KAP1 degradation model combined with the ectopic expression of KAP1 or its mutants to affirm previous findings that this protein serves as a transcriptional activator of HIV-1. Rather than using any of the available cellular models of latent HIV-1, the authors have opted to utilize HCT116 cells transiently transfected with a HIV LTR luciferase reporter since their diploid nature makes them amenable to the chemical genetic depletion of KAP1. Depletion of endogenous KAP1 in the HCT116 cells suppresses the stimulus-dependent activation of HIV but not basal promoter activity. However, the ectopic transient expression of KAP1 in these cells or three other cell lines is shown to be sufficient to elevate HIV basal promoter activity. Using the transient HIV reporter system, mutational analysis of KAP1 structural domains reveals previously unreported domains and sites that are functionally important in mediating the HIV-1 activator role of KAP1. Finally, using minimal promoter constructs containing specific individual cis-elements, the authors demonstrate that the RING domain-dependent transcriptional activity of KAP1 is not limited to the HIV-1 promoter and is not regulated by extracellularly-induced signaling mechanisms.

It is this reviewer’s impression that the study was well-conducted, the experiments presented have the proper technical controls and the manuscript for the most part has been well-written. The FKBP12-dTAG genetic model for controlling the depletion of KAP1 is a highly innovative aspect of this study and the authors should also be commended for extending their biochemical analysis to the use of site-directed mutations to bolster their observations with the deletion mutants.

This reviewer’s major point of criticism is that in attempting to validate that KAP1 is a positive regulator of HIV-1 during the ‘host phase’ of transcription, this study has solely relied on a transiently transfected reporter model. It is therefore likely that the luciferase reporter activity being assessed is emanating from episomal DNA rather than a stably integrated LTR reporter gene – a case in point is Figure 4a where all of the constructs employed in this experiment are likely being expressed in an episomal fashion. Also, since the HIV-1 LTR is unlikely to have been transcriptionally silenced over the 24-h period of transfection, one may wonder if there is a lingering promoter function that could be boosting the activity of KAP1. So the question still remains as to whether KAP1 can contribute to the signal-dependent reactivation of a latent, epigenetically silenced HIV-1.

Major comments:

1.    The authors ought to address the possibility that the transient transfection of the HIV-1 LTR luciferase reporter is resulting in its episomal expression and that this may confound the investigation of the role of KAP1 as a transcriptional activator of proviral HIV. Would it be possible to demonstrate in a bona fide latent HIV cell line model and using the FKBP12-dTAG approach that endogenous KAP1 indeed mediates a signal-induced reactivation of latent HIV? Otherwise, if this is not feasible, can the ectopic overexpression of KAP1 spontaneously reactivate HIV in latently infected cells?

2.    In Figure S1, why is the overexpression of KAP1 in HCT116 cells able to elevate HIV-1 LTR activity more robustly than its overexpression in the other three cell lines? If the latter cell lines are responsive to the stimuli used in the study (TNF, PMA), might there be an enhanced HIV-1 LTR activity in the KAP1-overexpressing cells due to stimulation similar to that observed in Figure 1e?

3.    While investigating the effects of KAP1 mutations in Figures 2, 3 and 4, those experiments were conducted with unstimulated cells. The observed repressive effects of certain mutants on the basal reporter activity are quite compelling and one would wonder what the results would look like in a stimulated condition if indeed KAP1 mediates signal-induced HIV-1 transcription. For instance, is it the case that if the RING domain of KAP1 has a more generalized activator role as indicated in Figure 4a, then HIV-1 reporter activity in the RING mutant cells should be less reponsive to the extracellular stimulus? Also, in reference to Figure S1 and lines 293-295, to what extent does the overexpression of KAP1 void the need for a stimulus?

4.    Since KAP1 appears to be constitutively expressed as seen in the DMSO control cells in Figure 1b and is resident in the nucleus, could the authors comment on how cellular stimulation is promoting the activity of KAP1 at the HIV promoter?

Minor comments:

1.    A number of syntactical and grammatical errors are present in the text: See lines 117-119, 150-151, 171, 174, 218. The sentence in lines 219-222 is somewhat confusing and may need to be rephrased. Line 95, “...for a different story currently under review” could also be rephrased.

2.    In the description in lines 509-519 the authors acknowledge the possibility that KAP1 may actually exercise two opposing functions, with a better functional characterization of its structural domains associated with the transcriptional repression of retroviruses. Could the findings from the current study be interpreted to suggest that KAP1 has a more dominant role as a transcriptional activator in the context of HIV? This seems to be relayed by the following statement in lines 428-432, “... our data is line with a large body of work in which KAP1 activates integrated proviruses in both transformed and primary CD4+ T cell models of latency [26, 27], strongly indicating KAP1 is required for signal-induced transcriptional activation and thus latency reactivation, but not latency maintenance [33, 34].”

3.    Similarly as the point above the statement in lines 433-435 indicating that KAP1 can function in a signal-independent manner is not consistent with the interpretation of the results in Figures 1c and 1e showing signal-induced KAP1 activity. This needs to be clarified.          

Comments on the Quality of English Language

Please see minor comments above.

Author Response

The responses are attached as PDF

Reviewer 3 Report

Comments and Suggestions for Authors

this paper demonstrated that KAP1 activates HIV-1 transcription as identified by a chemical genetics strategy, and further explored the mechanisms. this paper is well-written with clear logic.

questions:

1. is there any way to show that KAP1 plays an important role in HIV transcription in primary cells, including CD4 T cells and macrophages? 2. does antiretroviral therapy affect KAP1 function on HIV transcription? 3. does KAP1 expression differ in blood and tissue cells?

Comments on the Quality of English Language

this paper is well-written with clear logic

Author Response

The responses are attached as PDF
